# Habitual Videogame Playing Does Not Compromise Subjective Sleep Quality and Is Associated with Improved Daytime Functioning

**DOI:** 10.3390/brainsci13020279

**Published:** 2023-02-07

**Authors:** Oreste De Rosa, Francesca Conte, Paolo D’Onofrio, Serena Malloggi, Anna Alterio, Marissa Lynn Rescott, Fiorenza Giganti, Gianluca Ficca

**Affiliations:** 1Department of Psychology, University of Campania “L. Vanvitelli”, 81100 Caserta, Italy; 2Department of Psychology, University of Stockholm, 114 19 Stockholm, Sweden; 3Department NEUROFARBA, University of Florence, 50121 Florence, Italy

**Keywords:** videogames, sleep, sleep quality, sleepiness, circadian preference, depression, anxiety, stress

## Abstract

Research on the effects of videogames (VGs) on health has produced mixed results. Here, we assess the relationships of VG playing with sleep; chronotype; sleepiness; and levels of depression, anxiety, and stress; and how they are modulated by the level of exposure to VGs. Four hundred-and two adult participants (age = 26.2 ± 7.84; 227 F) completed an online survey including questions on VG use and a set of standardized questionnaires. The sample was divided into three groups: habitual gamers (HGs, 42.2%), nonhabitual gamers (NHGs, 36.5%), and non-gamers (NGs, 21.3%). No between-group differences emerged in sleepiness (Epworth Sleepiness Scale) or Pittsburgh Sleep Quality Index measures except the sleep disturbances subscore, which was higher in NHGs. HGs showed delayed bed- and risetimes and higher eveningness (reduced Morningness–Eveningness Questionnaire). HGs and NHGs showed higher depression subscores (Depression Anxiety Stress Scale) but remained in the subclinical range. Moreover, hours/week of VG playing predicted delayed sleep timing, lower daytime dysfunction, and lower sleepiness. Our data suggest that VG playing does not necessarily compromise sleep quality and may even benefit daytime functioning, underlining the need to reconsider the relationships between VG use and health by taking into account possible modulating factors such as habitual VG exposure.

## 1. Introduction

It has been a while since the first video game (from here on we will refer to the term as VG), a basic form of the famous game Tris, was created by Alexander S. Douglas in 1952. Since then, the impact and popularity of VGs have grown exponentially to make for one of the most profitable multimedia industries. The Entertainment Software Association (ESA) reported that over 226 million United States citizens, with an average age of 31 years, play VGs [1]. In 2021, it was reported that 35% of the Italian population aged 6–64 played VGs, with a peak in the 15–34 age range [2]. According to this report, people consider VGs as a source of relaxation, comfort, positive emotions, and a way to stay connected with others. Furthermore, playing VGs appears to be a social timekeeper capable of putting a line between work and free time [3].

In addition to the traditional use of VGs, there is a growing trend in recent years to watch videogame streaming (i.e., the user watches someone else playing through a specific online platform). According to the data report by Stream Hatchet [4], the first quartile of 2022 has recorded 8.8 billion VG stream-watchers worldwide. Given this, it is not surprising how important VGs have become in everyday life, especially at younger ages, and this has prompted researchers to focus on their impact from several aspects of psychological wellbeing, including sleep and sleepiness.

For years, VGs have mostly been related with adverse psychological and behavioral outcomes, especially with violent and antisocial behavior [5], and even recently, an APA document indicated a possible role of VGs (especially violent ones) in increasing aggressive behavior [6]. Indeed, several studies also report an increase of impulsiveness [7], anger [8], depression, and anxiety [9] linked to VG exposure. Impairments of sleep have also been repeatedly highlighted as a negative consequence of VG playing [10]. Specifically, studies assessing the impact of VGs on subsequent sleep found an increase in sleep onset latency [11], a reduction in total sleep time [12], and worsened sleep quality [11,13]. Moreover, VG playing was shown to be correlated with delayed average bedtime and risetime [14], reduced time in bed [15,16], and total sleep time [17], as well as with poor subjective sleep quality [14].

At first glance, these studies could crop up a rather infamous picture of VGs, but we believe that other less obvious aspects should be taken into account to look at this topic from a wider perspective. For instance, a recent review on the psychological impact of commercial VGs [18] underlined that VGs could be positively used to treat a large variety of mental conditions (e.g., mood disorders, PTSD, neurodevelopmental disorders, dementia, etc.), provided that several aspects of the intervention are controlled (e.g., type, frequency, duration of VG sessions, etc.). In line with this, many studies have observed positive effects of VGs, especially on emotion regulation (see [19] for a review). Specifically, not only have VGs repeatedly proven capable of decreasing depression, anxiety, stress [20,21,22,23], and frustration [24], but also of improving empathy [25]. Furthermore, both commercial and casual VGs have successfully been used to mitigate or treat psychological disorders [26,27,28]. Interestingly, Ferguson et al., who had previously underlined that the APA report on VGs [6] is incomplete since it overlooks quite a lot of studies [29], showed, in two experiments [30], that even violent VGs are unrelated to aggressive affect and behaviors expressed in the real world (independently of the platform used, i.e., traditional vs. virtual reality). Finally, VGs have been related to flourishing mental health and wellbeing (see [31] for a review).

As for sleep, our own group, in the frame of a research line dealing with the role of wake characteristics in determining sleep quality, has recently shown a positive impact of intense cognitive training on sleep architecture of a subsequent daytime nap [32] and of a subsequent night sleep episode [33] in terms of sleep continuity, stability, and organization. Since intensive training consisted of playing a bedtime game session (approximately 40 min) of a modified version of the VG Ruzzle, a positive effect of VGs on sleep was indirectly shown.

Furthermore, Ivarsson et al. conducted two studies on adolescents to understand the effects of violent (vs. nonviolent) VGs on sleep and heart rate (HR). In the first study [34], after exposure to a violent VG, participants reported higher activity in the very low frequency band of HR variability, but no effects on sleep (assessed through sleep diaries). In the second study [35], comparing high-exposed with low-exposed gamers (i.e., with the habit of playing 1 h or less daily), the authors found that the “violent condition” increased HR during sleep only for the low-exposed participants. Finally, through a questionnaire study, Altintas and colleagues [36] found that only the “intensity” of VG playing (a measure of addictive and compulsive VG use), and not the duration of VG sessions, correlated with higher scores on the Sleep Latency and Sleep Disturbances subscales of the Pittsburgh Sleep Quality Index [37] and was a more salient predictor of overall poor sleep quality. Finally, while it has been shown that VGs are negatively associated with morningness [38,39], findings are less clear about a possible VG–daytime sleepiness relationship. Some studies pointed out a negative impact of VGs [14,16] on vigilance, whereas Fossum and colleagues [38] found no significant association between VG use at bedtime and daytime sleepiness and Alsaad and colleagues [40] even found that attention was higher in expert vs. nonexpert gamers. Moreover, the latter study showed that expert and nonexpert gamers did not differ in habitual sleepiness or anxiety levels [40]. In line with these results, a recent meta-analysis reported that videogaming was not significantly associated with sleep quality [41].

These studies suggest that the numerous data on the negative effects of VGs could be due to several factors that have never been systematically investigated and probably deserve closer evaluation, such as the type of VG used (e.g., violent vs. nonviolent, commercial vs. casual, etc.), the subjective vs. objective nature of psychological measures, and the participants’ degree of practice with VGs, the time schedule and duration of VG sessions.

In this study, building on previous research [35,36,40], we focus on the habitual level of exposure to VGs as a possible modulating factor in the relationship of VG playing with sleep quality; chronotype; daytime sleepiness; and levels of depression, anxiety, and stress. Specifically, we administered, to a wide sample of Italian adults, an online survey including questions on VG playing habits as well as a set of standardized questionnaires on sleep, chronotype, sleepiness, depression, anxiety, and stress. In line with previous studies [35,40], we divided the sample according to the participants’ habitual level of exposure to VGs and compared psychological measures between the groups [35,40]. However, while those studies used a binary classification (high vs. low habitual gaming), we sought to better depict people’s VG habits by using three groups: habitual expert gamers (playing 7 or more hours/week), nonhabitual gamers (i.e., individuals who regularly play VGs but not more than 7 h/week), non-gamers (i.e., individuals who virtually never play VGs).

Our main working hypothesis is that habitual subjective sleep quality and daytime vigilance are not impaired in gamers relative to non-gamers, especially for habitual gamers, whose familiarity with gaming allegedly ends up preventing the increase in psychophysiological arousal.

An additional novelty of this study is the introduction of VG stream-watching as a behavior (additionally and related to VG playing) also potentially affecting psychological wellbeing. Given the growing diffusion of VG stream-watching, we aimed to assess its frequency in the population as well as its potential relationships with sleep, chronotype, sleepiness, depression, anxiety, and stress.

## 2. Materials and Methods

### 2.1. Participants and Procedure

Participants completed an anonymous online survey, which was distributed in Italy via university websites and social media between 1 September 2021 and 1 January 2022. Before starting the actual survey, participants were asked to read and fill an informed consent form, describing the study’s aims and methods in detail.

The only inclusion criterion was age ≥18.

The survey included sociodemographic questions (age, gender, employment status), a set of specific questions regarding videogaming and watching VG streaming and the following questionnaires in the validated Italian version: Pittsburgh Sleep Quality Index (PSQI [42]); Reduced Morningness–Eveningness Questionnaire (rMEQ [43]); Epworth Sleepiness Scale (ESS [44]); Depression Anxiety Stress Scales (DASS-21 [45]). The survey lasted approximately 20 min. No money or credit compensation was provided for participating in the study.

The study protocol was approved by the local ethical committee (code: 15/2021) and was conducted in accordance with the Declaration of Helsinki.

### 2.2. The Survey

After the first items on sociodemographic characteristics, three questions on videogaming and three on VG stream-watching were presented: (a) “On average, how many hours per day do you spend playing/stream-watching VGs?” (open question); (b) “On average, how many days per week do you play/stream-watch VGs?” (open question); (c) “At what time(s) of day do you typically play/stream-watch VGs?” (with 5 alternatives: “in the morning (before 12:00 p.m.”, “in the afternoon (12:00–06:00 p.m.)”, “in the evening (06:00–11:00 p.m.)”, “during the night (after 11:00 p.m.)”). The participant could select more than one answer to this item: for data analysis, these cases were classified as “at variable times”.

The first and second question on VG playing were used to classify the participants into three groups (average number of hours per day x average number of days per week): (a) habitual gamers (HGs), playing for 7 or more hours a week; (b) nonhabitual gamers (NHGs), playing <7 and >1 h a week; (c) non-gamers (NGs), playing 1 h or less a week.

Sleep timing and quality were assessed using the Italian version of the PSQI [42]. The scoring ranged from 0 to 21, with scores ≥5 indicating poor sleep quality [37]. The PSQI provides a global score and seven subscales: subjective sleep quality, sleep latency, sleep duration, habitual sleep efficiency, sleep disturbances, use of sleeping medication, and daytime disfunction over the last month; in addition, we also derived bedtime (hh:mm), risetime (hh.mm), time in bed (h), total sleep time (h), sleep onset latency (min), and sleep efficiency (%). PSQI has proven to demonstrate high efficiency in assessing sleep quality and differentiating between normal and pathological samples, with an overall reliability coefficient (Cronbach’s α) of 0.83.

Habitual levels of daytime sleepiness over the previous month were assessed using the ESS [44], an 8-item questionnaire requiring respondents to evaluate the probability of dozing off/falling asleep in different situations on a 4-point scale from 0 (“would never doze”) to 3 (“high chance of dozing”). The scoring ranges from 0 to 24, with scores >10 indicating the presence of a clinical condition of excessive sleepiness. ESS has demonstrated its usefulness in excessive daytime sleepiness evaluation, with moderate correlation with the Multiple Sleep Latency Test (MSLT).

Circadian preference was investigated using the r-MEQ [43], a five-item questionnaire, which classifies individuals in one of three chronotypes: evening type (scores 4–10), neither type (11–18), and morning type (19–25). Participants were asked to respond a set of questions describing the usual rhythm of sleep/wake activity. R-MEQ showed good external validity and strong correlations with actigraphic monitoring of activity.

Finally, we used the DASS-21 [45] to investigate depression, anxiety, and stress levels over the last 7 days. It is a self-report scale consisting of 21 items (7 for each subscale) based on a 3-point scale from 0 (“did not apply to me at all”) to 3 (“applied to me very much, or most of the time”). It provides a global score, by summing all the items, and a subscore for each subscale: depression (items: 3, 5, 10, 13, 16, 17, 21), anxiety (items: 2, 4, 7, 9, 15, 19, 20), and stress (items: 1, 6, 8, 11, 12, 14, 18). Cutoffs for the “normal range” at the different subscales are ≤9 for depression, ≤7 for anxiety, and ≤14 for stress [46]. DASS-21 has proven useful for assessing general distress and three dimensions (depression, anxiety, stress). Each dimension strongly correlates with measures of similar constructs and, combined, have an overall reliability coefficient (Cronbach’s α) of >0.70.

### 2.3. Statistical Analysis

Between-group differences in age, videogaming habits, sleep, chronotype, sleepiness, and psychological characteristics were assessed with the nonparametric Kruskal–Wallis ANOVA, due to the nonnormal distribution of variables (evaluated using the Shapiro–Wilk test). The Dwass–Steel–Critchlow–Fligner test (DSCF) was used for the post-hoc analysis. Difference in gender distribution was tested using the independent chi-square (χ2) test.

Predictors of sleep, chronotype, sleepiness, depression, anxiety, and stress were assessed using several multiple linear regressions. Specifically, we used hours/week spent playing videogames and hours/week spent stream-watching as predictors. For each overall model, we reported the F-, *p*-, and adjusted R2 values, and the standardized (β) coefficient and specific *p*-values for each predictor. Similarly, we used multiple linear regression analysis to test habitual time schedules of VG playing and VG stream-watching (morning, afternoon, evening, night, and variable times) as predictors of the same dependent variables.

Descriptive data are reported as mean ± standard deviation. Statistical significance was set at *p* ≤ 0.05. All analyses were conducted with Jamovi 2.0.0.0 [47].

## 3. Results

### 3.1. Descriptive Statistics

Out of the 402 participants (age = 26.3 ± 7.85 years, 227 F, all Italians and having Italian as mother tongue), 158 were habitual gamers (HGs: age = 25.6 ± 7, 44 F, hours/day playing VGs: 3.33 ± 1.81, days/week playing VGs: 5.55 ± 1.38, hours/week playing VGs: 18.8 ± 12.5), 138 were nonhabitual gamers (NHG: age = 26.6 ± 8.46, 97 F, hours/day playing VGs: 1.51 ± 0.78, days/week playing VGs: 2.23 ± 1.37, hours/week playing VGs: 3.07 ± 1.74), and 106 were non-gamers (NGs: age = 27.3 ± 8.19, 86 F, hours/day playing VGs: 0.76 ± 1.50, days/week playing VGs: 0.10 ± 0.38, hours/week playing VGs: 0.01 ± 0.01).

Groups did not differ in age (χ2 = 4.53, *p* = 0.104), whereas gender was not equally distributed (χ2 = 89.38, *p* ≤ 0.001).

Furthermore, 41.4% of participants reported to play at variable times during the day, 31.5% during the night, 14.2% in the afternoon, 11.7% in the evening, and only 1.2% in the morning.

### 3.2. Differences between HGs, NHGs, and NGs

#### 3.2.1. VG Stream-Watching

Table 1 reports comparisons of time spent VG stream-watching across the three groups. The profile of stream-watching was overall in line with that of VG playing, with more frequent weekly stream-watching in HGs compared to NHGs and NGs and in NHGs compared to NGs (all p’s < 0.01). Similarly, HGs and NHGs spent more hours/day stream-watching than NGs (both p’s < 0.01), whereas the difference between HGs and NHGs was not significant (*p* = 0.750).

#### 3.2.2. Sleep Variables, PSQI Global Score, and PSQI Subscores

Table 2 displays ANOVA results on sleep variables, PSQI global score, and PSQI subscores. Among the sleep variables, only bedtime and risetime significantly differed across groups, with HGs reporting delayed bedtime compared to NHGs (*p* = 0.050) and NGs (*p* ≤ 0.001) and delayed risetime compared to NGs (*p* = 0.007). PSQI global score did not differ across groups, nor did PSQI subscores, except for sleep disturbances: specifically, NHGs reported more sleep disturbances than HGs (*p* = 0.050) and NGs (*p* = 0.040). Finally, the three groups had similar proportions of good and poor sleepers.

#### 3.2.3. Chronotype, Sleepiness, and Psychological Variables

The analysis on rMEQ scores yielded a significant difference across groups (HGs: 14.4 ± 2.79, NHGs: 14.9 ± 2.92, NGs: 15.3 ± 2.54; χ2 = 6.96, *p* = 0.031), with HGs showing higher eveningness than NGs (*p* = 0.012); other pairwise comparisons were nonsignificant. Habitual sleepiness, instead, was similar across groups (HGs: 8.58 ± 5.03, NHGs: 8.94 ± 5.18, NGs: 8.67 ± 5.19; χ2 = 0.390, *p* = 0.823).

As for psychological measures (Figure 1), the DASS-21 global score did not differ across groups (HGs: 22.9 ± 13.5, NHGs: 23.3 ± 13.5, NGs: 20.0 ± 14.4; χ2 = 5.60, *p* = 0.061), nor did the DASS-21 anxiety subscale (HGs: 5.73 ± 4.38, NHGs: 6.09 ± 4.61, NGs: 5.03 ± 4.79; χ2 = 5.66, *p* = 0.059) or the stress subscale (HGs: 9.52 ± 5.10, NHGs: 10.4 ± 5.21, NGs: 9.62 ± 5.91; χ2 = 2.26, *p* = 0.322). Instead, a significant difference emerged for the depression subscale (HGs: 8.59 ± 6.02, NHGs: 7.62 ± 5.65, NGs: 5.93 ± 5.58; χ2 = 14.59, *p* < 0.001), with HGs and NHGs reporting higher scores than NGs.

### 3.3. Regression Analyses with Hours Spent VG Playing or VG Stream-Watching as Predictors

#### 3.3.1. Sleep Variables, PSQI, and PSQI Subscales

Results of regression analyses on sleep variables and PSQI scores are displayed in Table 3. As for sleep variables, the analysis yielded a significative regression model for sleep timing measures, with hours/week spent VG playing positively predicting delayed bed- and risetimes.

No significant regression model emerged for any PSQI variable except for the daytime dysfunction subscore. Specifically, hours/week spent playing VGs predicted lower scores at this subscale (indicating better daytime functioning).

#### 3.3.2. Chronotype, Sleepiness, and Psychological Variables

The linear regression on chronotype, sleepiness, and psychological measures (Table 4) yielded a significant effect only for sleepiness. Specifically, hours/week spent playing VGs predicted lower levels of habitual sleepiness.

### 3.4. Regression Analyses with Schedules of VG Playing and VG Stream-Watching as Predictors (HGs and NHGs)

VG playing schedules did not significantly predict any sleep, sleepiness, or chronotype variable, except for risetime (F4,319 = 3.17, R2 = 0.026, *p* = 0.014), which, unsurprisingly, was predicted by afternoon (β = 1.02, *p* = 0.047), night (β = 1.32, *p* = 0.009), and variable schedules (β = 1.21, *p* = 0.016) relative to a morning schedule. More interestingly, VG playing in the afternoon (β = 1.16, *p* = 0.026), night (β = 0.1.08, *p* = 0.033), and at variable times (β = 1.18, *p* = 0.019) relative to the morning positively predicted the DASS-21 global score (F4,319 = 2.31, R2 = 0.028, *p* = 0.050). At the same time, playing in the afternoon (β = 1.09, *p* = 0.034), night (β = 1.12, *p* = 0.027), and at variable times (β = 1.31, *p* = 0.009) compared to the morning positively predicted scores in the depression subscale (F4,319 = 2.68, R2 = 0.032, *p* = 0.032). There was no significant predictor for stress and anxiety subscales.

The results of VG stream-watching schedules were very similar. VG stream-watching at night significantly predicted delayed bedtime (F4,319 = 3.43, R2 = 0.041, *p* = 0.009) relative to afternoon (β = −0.42, *p* = 0.016) and evening (β = −0.59, *p* = 0.002), and delayed risetime (F4,319 = 3.17, R2 = 0.026, *p* = 0.014) relative to morning (β = −1.32, *p* = 0.009) and evening (β = −0.44, *p* = 0.018). Night schedules also predicted higher scores on the DASS-21 depression subscale relative to morning schedules (F4,319 = 2.68, R2 = 0.032, *p* = 0.032; β = −1.12, *p* = 0.027). No other significant predictor emerged.

## 4. Discussion

With this survey study, we aimed to investigate the relationships of VG playing with sleep, chronotype, daytime sleepiness, and levels of depression, anxiety, and stress, as well as how these relationships are modulated by levels of habitual exposure to VGs. To this aim, based on the frequency of VG playing and on the average time spent engaging in this activity, we identified three groups of participants: habitual gamers (HGs), nonhabitual gamers (NHGs), and non-gamers (NGs).

Taking into account the amount of exposure to VGs appears very important in any investigation on VGs and wellbeing. In fact, as shown by Ivarsson and colleagues [35], VG playing could have different effects, not necessarily negative, depending on the extent to which people engage in this activity. In line with this, our main result is the absence of any negative effect of VG playing on sleep. In fact, between-group comparisons on sleep variables revealed no differences except for bed- and risetimes (which were delayed in HGs). This suggests that, despite delayed sleep timing, HGs’ overall sleep architecture is basically unaltered. Moreover, HGs’ subjective sleep quality and daytime alertness are also unaffected, as shown by the finding that the global PSQI score and the proportion of good and poor sleepers, as well as habitual sleepiness levels, were similar across groups.

An analogous pattern emerged for all PSQI subscales except “sleep disturbances”, which was higher in NHGs compared to the other two groups, in agreement with Ivarsson and colleagues’ finding [35] of an HR increase during sleep only in the “low exposition” (to VGs) group. This result could possibly be explained by an increase in psychophysiological arousal levels caused by occasional videogaming, which results in subsequent sleep impairment; instead, habituation to gaming in HGs prevents these changes in arousal level.

Our observation of delayed bed- and risetimes in HGs is in line with the finding that hours/week of VG exposure predict delayed bed- and risetimes and is coherent with the greater evening preference reported by HGs. These results are consistent with several previous studies showing associations of VG exposure to delayed sleep timing and eveningness [14,38,39]. Indeed, unsurprisingly, people generally use VGs in leisure time, typically corresponding to later times of day. In line with this, only 1.2% of our participants reported playing VGs in the morning. However, based on our own and previous data, the question remains open on the direction of the relationship between habitual gaming and evening preference, although a bidirectional influence is most likely.

Our results from regression analyses on sleep and PSQI measures further support the idea that VG exposure, even at high levels as in HGs, does not necessarily hinder sleep quality and daytime functioning, which is consistent with a recent meta-analysis [41]. In fact, with the exception of bed- and risetimes, neither hours of VG playing nor of VG stream-watching emerged as significant predictors of any sleep measure. Similarly, they did not predict global PSQI score nor PSQI subscores, except for the daytime dysfunction subscore, which actually showed a negative relationship with hours of VG playing. In other words, more hours of VG playing is associated with lower daytime dysfunction. Also coherent with this result is the finding that hours of VG playing significantly predicted lower sleepiness levels. These data resemble those of Alsaad and colleagues [40], who found beneficial effects of VG playing on vigilance and attention, and of previous research from our group, showing that polysomnographically monitored sleep after an intensive VG session was even improved in terms of sleep continuity, stability, and cyclic organization [32,33]. Furthermore, the observed positive association between hours of VG playing and daytime functioning is consistent with research from the sleep–memory field, showing that performance at cognitive tasks administered at bedtime (often in the form of VGs) is enhanced after the sleep episode [48,49].

Instead, our data are in contrast with Exelman and van den Bulck’s findings [14,50] of a negative association between videogaming duration and frequency with subjective sleep quality (measured through the PSQI, as in this study). However, it is important to note that the authors did not consider the participants’ levels of experience with VGs, i.e., their analyses were conducted on the total sample with no distinctions made according to videogaming habits. Therefore, it cannot be excluded that their participants were mostly only occasional gamers, a hypothesis supported by the fact that, in the first study [14], the average amount of time spent VG playing by participants was 22.87 min/day (i.e., even less than the average time of VG playing reported by our sample of NGs).

Turning to psychological measures, gamers (both HGs and NHGs) reported higher depression levels than NGs, but notably, their scores remained in the “normal” range [46]. Moreover, hours/week spent VG playing or stream-watching were not related to either the DASS-21 global score or any of its subscores. Therefore, it appears that habitual and occasional VG playing bears some relation to subclinical depressive symptoms regardless of the amount of time spent playing. This finding cannot be interpreted univocally. In fact, in relation to depressive symptoms, VG playing could be hypothesized to be (a) a determinant or maintaining factor (e.g., the desire to prolong VG sessions would entail withdrawal from other more social activities), (b) a consequence (e.g., withdrawal from outdoor and social activities could lead to preferring more solitary ones such as VG playing), (c) a coping strategy (e.g., VG playing would allow an individual to find relief from painful emotions). The first hypothesis appears unlikely, since NHGs (rather than only HGs) also showed higher scores on the depression subscale and because we did not find any significant association between time spent playing and the depression subscore. In other words, it is unlikely that very few hours per week spent playing VGs, as in the NHG sample, would be enough to determine a significant withdrawal from other activities; moreover, a relation between VG exposure time and depression would have been expected. Based on our data, the other two hypotheses appear more plausible. Indeed, the recent literature, showing the benefit of VG playing on depression, anxiety, stress [20,21,22,23], and frustration [24], as well as its successful use in improving empathy [25] and mitigating psychological disorders [26,27,28], lends greater support to the idea that VG playing could be a positive coping strategy rather than a hindrance to psychological wellbeing. In line with this, an IDEA report [51] pointed out that, during the COVID-19 pandemic, VG use helped people cope with the unpleasant implications of the restrictions, to relax, and to remain connected with others.

As for VG stream-watching, the literature on the topic is very scarce and generally includes this activity in the wider framework of multimedia use and binge-viewing, with contrasting results on its effects on psychological wellbeing [52,53,54]. In this regard, our data support the hypothesis that VG stream-watching can be more specifically considered as a complementary activity to VG playing [52] since, in our sample, the profile of time spent stream-watching was similar to that of time spent playing VGs (HGs reported to stream-watch the most, followed by NHGs, and finally NGs). Moreover, we did not observe any association between time spent stream-watching and any sleep, sleep quality, circadian preference, sleepiness, or psychological variable, suggesting that, similar to VG playing, this activity does not yield negative effects on these aspects of wellbeing. Anyway, further research is needed to better describe the effects of VG stream-watching and possibly to disentangle them from those of other activities such as VG playing and binge-viewing.

A few limitations must be acknowledged, which impose caution in the interpretation of our findings. First, our results should be considered in light of methodological constraints linked to the nature of survey studies: for instance, we cannot exclude that responses were biased by recall accuracy of respondents or that the questionnaire might have preferentially attracted respondents in some way interested in the world of VGs. Moreover, the three groups differed in gender distribution: specifically, females were underrepresented in the HG group and overrepresented in the other two groups. However, this unequal distribution of VG exposure across genders was expected considering that (a) more men than women play VGs according to several studies and ESA reports (e.g., [1,55]); (b) women are more likely than men to underreport the amount of time they spend playing VGs [56].

## 5. Conclusions

In conclusion, our findings show that VG use in the general adult population is not detrimental to important aspects of wellbeing such as sleep quality and daytime functioning and is not associated with clinical levels of depression, anxiety, or stress. In addition, they suggest that VG playing could even benefit daytime functioning in terms of alertness and cognitive efficiency. Thus, our pattern of results encourages the reconsideration of the previous literature on the negative effects of VG use and prompts further research aimed at better clarifying the role of possible modulating factors such as the degree of practice, the violent vs. nonviolent nature of the VG in question, complexity of the VG, age, and associated psychopathological characteristics (e.g., use of VGs in an addictive or compulsive way). In this regard, it is worth noting, for instance, that most experimental studies documenting negative consequences of VG exposure [10] have used high-impact commercial VGs (i.e., with high action in first or third person and a number of instructions for playing and controlling the joystick), which are more likely to produce arousing effects. Along the same line, it is plausible that immoderate and compulsive VG use, especially during childhood and adolescence, can yield adverse effects on sleep and health [57]; however, general VG use, devoid of these characteristics, appears unassociated with negative consequences, as shown in this study, as well as in the previous literature (e.g., [37]). Furthermore, more in-depth investigations on how VG use affects sleep features and psychological wellbeing are required. Indeed, studies on VGs and sleep have mostly focused on traditional sleep variables (e.g., sleep onset latency, time in bed, total sleep time, sleep efficiency), which may be insufficient to capture more subtle changes in sleep dynamics. For instance, we have previously shown that several objective measures of sleep continuity, stability, and cyclic organization were improved after a bedtime VG playing session [32,33]. These data, along with the positive association between VG playing and daytime functioning observed here and with the extant literature on the positive effects of VGs on several psychological measures [19,31], suggest that a careful and controlled use of VGs may have significant applications as a low-cost and feasible strategy to promote health. Therefore, a clearer characterization of which modulating factors influence the effects of VGs on wellbeing and of the specific aspects of health that are involved appears all the more necessary.

## Figures and Tables

**Figure 1 brainsci-13-00279-f001:**
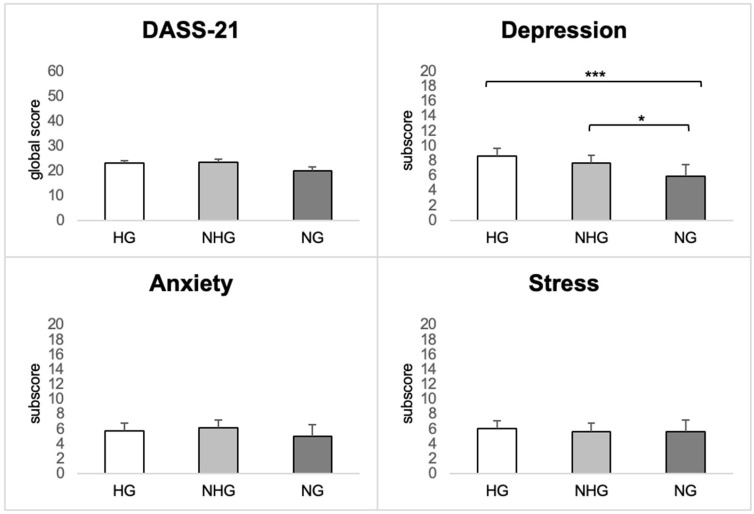
DASS-21 global score and DASS-21 subscores in the three groups. *** *p* < 0.001; * *p* < 0.05.

**Table 1 brainsci-13-00279-t001:** Frequency per week and hours per day of stream-watching in the three groups.

Variables	Groups	m ± sd	*χ* ^2^	*p*
Days/week stream-watching	HG	2.63 ± 1.91	31.7	**<0.001**
	NHG	1.78 ± 1.55		
	NG	0.63 ± 1.05		
Hours/day stream-watching	HG	1.27 ± 0.71	12.0	**0.002**
	NHG	1.18 ± 0.76		
	NG	0.57 ± 0.64		

Note: significant differences are in bold.

**Table 2 brainsci-13-00279-t002:** Sleep variables, PSQI global score, and PSQI subscores in the three groups.

Variables	Groups	m ± sd	*χ* ^2^	*p*
Bedtime (hh:mm)	HG	00:46 ± 01:32	14.70	**<0.001**
	NHG	00:22 ± 01:19		
	NG	00:16 ± 01:16		
Risetime (hh:mm)	HG	08:43 ± 01:48	9.98	**0.007**
	NHG	08:25 ± 01:24		
	NG	08:18 ± 01:12		
SOL (min)	HG	26.2 ± 27.4	0.87	0.645
	NHG	26.0 ± 27.8		
	NG	22.5 ± 19.1		
TIB (h)	HG	8.02 ± 1.27	0.32	0.849
	NHG	7.97 ± 1.26		
	NG	8.07 ± 1.27		
TST (h)	HG	7.04 ± 1.34	0.00	0.999
	NHG	7.04 ± 1.49		
	NG	6.99 ± 1.23		
Efficiency (%)	HG	88.5 ± 13.6	1.21	0.545
	NHG	88.8 ± 15.7		
	NG	88.4 ± 18.5		
PSQI global score	HG	6.46 ± 3.14	0.47	0.787
	NHG	6.66 ± 2.98		
	NG	6.66 ± 3.17		
Subjective sleep quality subscore	HG	1.30 ± 0.66	0.94	0.624
	NHG	1.27 ± 0.66		
	NG	1.24 ± 0.57		
Sleep latency subscore	HG	1.26 ± 1.03	2.10	0.350
	NHG	1.21 ± 0.90		
	NG	1.08 ± 0.96		
Sleep duration subscore	HG	1.04 ± 0.95	0.02	0.988
	NHG	1.06 ± 1.02		
	NG	1.08 ± 1.05		
Sleep efficiency subscore	HG	0.66 ± 1.02	3.86	0.144
	NHG	0.68 ± 1.05		
	NG	0.87 ± 1–12		
Sleep disturbances subscore	HG	1.19 ± 0.51	7.87	**0.019**
	NHG	1.33 ± 0.54		
	NG	1.18 ± 0.51		
Sleep medication subscore	HG	0.17 ± 0.61	0.31	0.855
	NHG	0.13 ± 0.52		
	NG	0.23 ± 0.77		
Daytime dysfunction subscore	HG	0.83 ± 0.66	4.88	0.087
	NHG	0.97 ± 0.58		
	NG	0.92 ± 0.70		
Number of good sleepers (GS) and poor sleepers (PS)	HG	75 GS, 83 PS	1.74	0.419
	NHG	55 GS, 83 PS		
	NG	47 GS, 59 PS		

Note: significant differences are in bold.

**Table 3 brainsci-13-00279-t003:** Linear regression results for sleep variables and PSQI scores.

Predictors	F_2,389_	*p* (Overall)	Adj. R^2^	β	*p*	Variables
VG playing	8.53	**<0.001**	0.037	0.20	**<0.001**	Bedtime (hh:mm)
VG stream-watching				0.02	0.741	
VG playing	10.6	**<0.001**	0.047	0.23	**<0.001**	Risetime (hh:mm)
VG stream-watching				–0.02	0.648	
VG playing	0.20	0.817	–0.004	0.03	0.540	SOL (min)
VG stream-watching				–0.02	0.738	
VG playing	0.84	0.430	–0.004	0.06	0.251	TIB (h)
VG stream-watching				–0.05	0.366	
VG playing	0.48	0.615	–0.003	0.05	0.373	TST (h)
VG stream-watching				0.01	0.875	
VG playing	0.31	0.730	–0.004	–0.01	0.774	Efficiency (%)
VG stream-watching				0.04	0.430	
VG playing	1.39	0.251	0.002	–0.09	0.106	PSQI tot
VG stream-watching				0.04	0.410	
VG playing	0.09	0.914	0.004	–0.21	0.679	Subjective sleep quality
VG stream-watching				0.01	0.840	
VG playing	1.06	0.346	0.004	0.02	0.665	Sleep latency
VG stream-watching				0.06	0.224	
VG playing	0.60	0.550	–0.002	–0.06	0.290	Sleep duration
VG stream-watching				0.03	0.577	
VG playing	1.66	0.191	0.003	–0.10	0.069	Sleep efficiency
VG stream-watching				0.02	0.685	
VG playing	1.16	0.314	0.004	–0.08	0.131	Sleep disturbances
VG stream-watching				0.01	0.816	
VG playing	0.05	0.956	–0.004	0.01	0.796	Sleep medication
VG stream-watching				0.00	0.938	
VG playing	2.90	**0.050**	0.001	–0.12	**0.021**	Daytime dysfunction
VG stream-watching				0.00	0.931	

Note: significant differences are in bold.

**Table 4 brainsci-13-00279-t004:** Linear regression results for sleep variables and PSQI scores.

Predictors	F_2,389_	*p* (Overall)	Adj. R^2^	β	*p*	Variables
VG playing	1.31	0.272	0.007	–0.03	0.583	rMEQ
VG stream-watching				–0.07	0.192	
VG playing	4.50	**0.012**	0.018	–0.15	**0.005**	ESS
VG stream-watching				0.10	0.085	
VG playing	0.18	0.837	–0.004	–0.01	0.883	DASS-21
VG stream-watching				0.03	0.552	
VG playing	2.37	0.095	0.007	0.09	0.073	Depression
VG stream-watching				0.04	0.497	
VG playing	0.23	0.797	–0.004	–0.02	0.660	Anxiety
VG stream-watching				0.03	0.540	
VG playing	1.44	0.238	0.002	–0.09	0.091	Stress
VG stream-watching				0.02	0.712	

Note: significant differences are in bold.

## Data Availability

The data presented in this study are available on request from the corresponding author. The data are not publicly available due to privacy reasons.

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
