# Peer review of "Habitual Videogame Playing Does Not Compromise Subjective Sleep Quality and Is Associated with Improved Daytime Functioning"

_brainsci, 2023, doi:10.3390/brainsci13020279_

Round 1

Reviewer 1 Report

The subject of this study is very interesting and topical. However, I have a few comments. The originality of the study and the scales of the survey are not sufficiently developed. 

The title of the article can be improved.

Please reformulate or develop your hypothesis. Oriented hypothesis is necessary. What is your prediction? Insist on the original contribution of the study.

What is the place of the circadian rhythm in this study? 

What is the relationship between sleep and psychological health? Why this double measure?

Please describe more deeply the classification in three groups and argument this method or choice.

Please indicate the maternal language of the sample. 

Please indicate the “number of approved” decision by the local ethical committee

Indicate if the different scales are validated in Italian and reported the psychometric qualities of each scale? Develop the description of the different scale assessment.

How to explain the he non-normal distribution of variables?

Please develop the limit of this study, the practical implication.

Author Response

The subject of this study is very interesting and topical. However, I have a few comments. The originality of the study and the scales of the survey are not sufficiently developed. 

Point 1: The title of the article can be improved.

Response 1: The title has been changed to include an explicit mention of the study’s main findings.

Point 2: Please reformulate or develop your hypothesis. Oriented hypothesis is necessary. What is your prediction? Insist on the original contribution of the study.

Response 2: The end of the Introduction section has been modified and enriched according to the Reviewer’s suggestions: specifically, the working hypothesis has been clarified and the originality of the study has been highlighted (line 122).

Point 3: What is the place of the circadian rhythm in this study?

Response 3: We have added regression analyses to assess the activity schedules predictors (for VG playing and VG watch-streaming: morning, afternoon, evening, night, variable) of sleep, chronotype, sleepiness and psychological variables. We had previously excluded these analyses from the final version of the manuscript because they are limited by two factors: 1) there are very few participants reporting to play in the morning (only 4); 2) about the 40% of participants selected more categories and were scored to play/watch at “variables times”.

Point 4: What is the relationship between sleep and psychological health? Why this double measure?

Response 4: We thank the Reviewer for noticing this inaccuracy. We acknowledge that “psychological health” is a broad concept which includes sleep health. Here our main focus is on sleep and sleep-related measures but we have also included an assessment of depression, anxiety and stress to investigate possible associations between VG exposure and other psychological indices of well-being. Moreover, given the strong links between sleep problems and these psychological factors, the latter were also considered as possible confounders (to be controlled for) of the relationships between VG exposure and sleep. Anyway, to avoid being misleading, we have deleted the expression “psychological health” throughout the manuscript.

Point 5: Please describe more deeply the classification in three groups and argument this method or choice.

Response 5: The classification in three groups as well as its reasons are now more clearly described at the end of the Introduction section (line 117).

Point 6: Please indicate the maternal language of the sample. 

Response 6: Done (line 140).

Point 7: Please indicate the “number of approved” decision by the local ethical committee.

Response 7: Done (line145).

Point 8: Indicate if the different scales are validated in Italian and reported the psychometric qualities of each scale? Develop the description of the different scale assessment.

Response 8: All the standardized questionnaires included in the survey were validated in Italian, as now specified in the text (line 135). Psychometric properties of each scale have been reported and the description of the different scales has been further clarified.   

Point 9: How to explain the non-normal distribution of variables?

Response 9: As described in literature it is not surprising that psychological variables move on a non-normal distribution (Bianca et al., 2013; Bono et al., 2017, review). We also found non-normal distribution of variables in a past work for some of the variables here assessed (e.g., PSQI variables, Conte et al., 2021;). Furthermore, although some authors consider acceptable levels of Skewness and Kurtosis between + 2 – 2 (see George & Mallery, 2010), as good practice, we typically use the more conservative Shapiro-Wilk test to assess the normality distribution of variables.

Point 10: Please develop the limit of this study, the practical implication.

Response 10: The limitations of the study and their implications have been described in an additional paragraph at the end of the Discussion section.

Reviewer 2 Report

Dear Authors, Thank you for the opportunity to read and evaluate this manuscript.

The study of the relationship between sleep and video games is not a new topic, although the perspective of streaming video games has seemed refreshing to me.

I consider that the article uses current references and the amount is justified and adequate. The results are clear and well written. 

Author Response

Dear Authors, Thank you for the opportunity to read and evaluate this manuscript.

The study of the relationship between sleep and video games is not a new topic, although the perspective of streaming video games has seemed refreshing to me.

I consider that the article uses current references and the amount is justified and adequate. The results are clear and well written.

We thank the reviewer.

Reviewer 3 Report

Dear Authors. The research is correct. However that the introduction and discussion could be strengthened by adding more meta-analysis studies and systematic reviews.

Author Response

Point 1. Dear Authors. The research is correct. However that the introduction and discussion could be strengthened by adding more meta-analysis studies and systematic reviews.

Response 1. We thank the reviewer for the comment. There are very few studies investigating the relationships between sleep/psychological health and video games in adults, however we cited more reviews as suggested (lines: 76, 103, 329, 426).